# Molecular Imaging of Central Dopamine in Obesity: A Qualitative Review across Substrates and Radiotracers

**DOI:** 10.3390/brainsci12040486

**Published:** 2022-04-08

**Authors:** Lieneke Katharina Janssen, Annette Horstmann

**Affiliations:** 1Department of Neurology, Max Planck Institute for Human Cognitive and Brain Sciences, 04103 Leipzig, Germany; horstmann@cbs.mpg.de; 2Institute of Psychology, Otto von Guericke University Magdeburg, 39106 Magdeburg, Germany; 3Department of Psychology and Logopedics, Faculty of Medicine, University of Helsinki, 00014 Helsinki, Finland

**Keywords:** dopamine, obesity, BMI, Positron Emission Tomography, single-photon emission tomography

## Abstract

Dopamine is a neurotransmitter that plays a crucial role in adaptive behavior. A wealth of studies suggests obesity-related alterations in the central dopamine system. The most direct evidence for such differences in humans comes from molecular neuroimaging studies using positron emission tomography (PET) and single-photon emission computed tomography (SPECT). The aim of the current review is to give a comprehensive overview of molecular neuroimaging studies that investigated the relation between BMI or weight status and any dopamine target in the striatal and midbrain regions of the human brain. A structured literature search was performed and a summary of the extracted findings are presented for each of the four available domains: (1) D2/D3 receptors, (2) dopamine release, (3) dopamine synthesis, and (4) dopamine transporters. Recent proposals of a nonlinear relationship between severity of obesity and dopamine imbalances are described while integrating findings within and across domains, after which limitations of the review are discussed. We conclude that despite many observed associations between obesity and substrates of the dopamine system in humans, it is unlikely that obesity can be traced back to a single dopaminergic cause or consequence. For effective personalized prevention and treatment of obesity, it will be crucial to identify possible dopamine (and non-dopamine) profiles and their functional characteristics.

## 1. Introduction

Endogenous dopamine enables signal transduction and modulation throughout the central nervous system, and contributes to synaptic plasticity. Dopamine plays a crucial role in adaptive behaviour, such as behavioural and cognitive control [1,2,3,4,5,6], reinforcement learning [7,8,9], and motivation [10,11,12,13]. Most relevant to these processes is dopamine’s action in mesolimbic regions of the brain—striatum, substantia nigra (SN) and ventral tegmental area (VTA)—and in the prefrontal cortex.

A wealth of evidence suggests obesity-related differences in neurocognitive measures of these dopamine-dependent processes in humans, reviewed in [14], indirectly pointing to a role for dopamine in the development and maintenance of diet-induced obesity. In the past decades, human obesity has been associated with a variety of changes in dopamine pathways in the central nervous system [14,15,16,17,18,19,20]. Dopamine alterations that result from chronic low-grade inflammation, as also observed in obesity [21], have recently been proposed as an explanation for an inability (rather than unwillingness) to put in the effort required to achieve a desired goal [22]. Although highlighting the functional relevance for (mal)adaptive behaviour, the exact nature of structural dopamine imbalances remains elusive despite research on the relationship between diet-induced obesity and the brain’s dopamine system being in full bloom. Given the importance of effortful lifestyle changes (i.e., in diet and physical activity) for the successful prevention and treatment of obesity, the central dopamine system presents a functionally relevant therapeutic target.

Structural dopamine alterations in human obesity have been demonstrated in molecular imaging studies, in particular in the domain of dopamine D2/D3 receptor availability and dopamine release, but dopamine synthesis capacity and dopamine reuptake transporters have also been under investigation. Positron emission tomography (PET) and single-photon emission computed tomography (SPECT) are the most direct ways to investigate dopaminergic changes in humans in vivo. Although dopamine levels cannot be measured directly, certain molecular substrates of the central dopamine system can be approximated by means of highly-selective radiolabeled tracers (Figure 1). Radiotracers bind with a certain affinity to a molecule that is part of the dopamine system. For most tracers, their binding potential is quantified by localizing the source and amount of the radioactivity they emit. The affinity of tracers is often regionally-specific and determines how reliably binding of the tracer in that and other regions of the brain can be quantified. Most dopamine radiotracers are optimal for quantifying alterations in the striatum, and sometimes in the midbrain regions SN and VTA too. Due to tremendous ethical and financial considerations, single dopamine PET and SPECT studies typically target only one substrate and they are performed with relatively small and limited samples in a cross-sectional design (with exceptions). This approach does not match the complex nature of the dopamine system nor of obesity etiology. Careful synthesis of the wealth of available molecular imaging findings is therefore required.

Existing molecular imaging findings have been discussed in a number of excellent reviews that focus on different subsets of the four available dopamine domains [15,16], [23,24,25,26,27]. The aim of the current review is to give a comprehensive overview of molecular neuroimaging studies that investigated the relation between BMI (on a continuum) or obesity (as a weight status) and any dopamine target in the striatal and midbrain regions of the human brain. First, a description of the structured literature search is provided, followed by a summary of the extracted findings for each of the four available domains: (1) D2/D3 receptors, (2) dopamine release, (3) dopamine synthesis, and (4) dopamine transporters. Recent proposals of a nonlinear relationship between severity of obesity and dopamine imbalances are then described while integrating findings within and across domains. Further, limitations of the review strategy and the reviewed studies are discussed, followed by a general conclusion. Throughout the text, gaps in the literature are identified that, when addressed, could help the field transform current speculation into theory.

## 2. Literature Search

A structured, qualitative review of the literature was performed by LKJ (Figure 2). To ensure a comprehensive review of the available studies, recommendations for the systematic review of non-intervention studies (NIRO-SR)[28] were followed as closely as possible. The initial PubMed search (https://pubmed.ncbi.nlm.nih.gov/ last accessed on 28 August 2021) was performed on 9 December 2020 using the search terms “((Positron Emission Tomography) OR (Single-Photon Emission Computerized Tomography)) AND ((Obesity) OR (BMI)) AND (dopamine)” and resulted in 91 hits. On 28 August 2021 the search was repeated to check for new publications. Despite six more hits, no new empirical articles were retrieved. Titles and abstracts of all 97 hits were screened. Only empirical PET and SPECT studies in human volunteers that reported striatal and/or midbrain measurements of at least one dopamine radiotracer were included, and statistically tested for a correlation with BMI and/or a group difference between BMI groups with no known eating disorders. After screening, the full texts and references of the remaining 31 articles were assessed in detail. One article did not include dopamine measurements in the striatum or midbrain and was excluded [29]. One relevant article was missed in the original search and was added, resulting in a total of 31 studies that covered BMI/obesity-related findings in the domains of D2/D3 receptors (*n* = 19), dopamine release (*n* = 6), dopamine synthesis (*n* = 3), and dopamine transporters (*n* = 7). Several articles reported multiple findings from different domains, either from primary or secondary analyses, as inferred from the objectives, aims, or hypotheses stated in the introduction. Findings from secondary analyses have also been reviewed to synthesize all available records. Table 1, Table 2, Table 3 and Table 4 report details on imaging and sample characteristics that were extracted manually along with the BMI/obesity-related findings for each domain separately.

## 3. Summary of the Findings

### 3.1. D2/D3 Receptors

Obesity has been associated with differences in striatal D2/D3 receptor binding potential in twelve studies using the SPECT-radiotracer [^123^I]-IBZM, or the PET-radiotracers [^11^C]-Raclopride, [^18^F]-Fallypride, or [^11^C]-(+)-PHNO (Table 1). Six studies indicated higher binding potential [40,41,42,43,44,46], whereas seven studies indicated lower binding potential [30,31,32,33,34,44,45]. However, in eight other studies using the PET-radiotracers [^11^C]-Raclopride or [^11^C]-NMB, no obesity-related differences were reported [35,36,37,38,39,40,47,48].

Closer inspection suggests that a lower striatal D2/D3 receptor binding potential may be characteristic of more severe obesity, here referred to as extreme obesity (discussed in Section 4.2). The studies that included individuals with a BMI above 55 kg/m^2^ and compared the obese group to a non-obese control group (BMI < 30 kg/m^2^) consistently found lower binding [30,31,32,34]. Secondary correlational analyses, however, do not point to a linear association between BMI and D2/D3 receptor binding potential in the obese group [30,31,34,36], except in the original study by Wang and colleagues [32].

For mild to severe obesity the evidence points towards higher striatal D2/D3 receptor binding potential, although the picture is less clear. A group study in obese women with a BMI below 55 kg/m^2^ shows higher D2/D3 receptor binding in the caudate nucleus relative to non-obese women, which was measured with [^18^F]-Fallypride [43]. Higher striatal binding has also been associated with higher BMI (upper BMI limit 35–40 kg/m^2^) in women and men in two correlational studies using the same radiotracer [44,46]. Furthermore, three studies with the radiotracer [^11^C]-(+)-PHNO consistently report higher striatal D2/D3 receptor binding with higher-BMI-up-to-severely obese [42], moderately obese [41], and non-obese [40] women and men. However, lower striatal binding has also been reported in two correlational studies with the radiotracer [^18^F]-Fallypride [44,45], and in a group study using [^11^C]-Raclopride [33]. Three further group studies found no difference in binding potential between obese (BMI < 55 kg/m^2^) and non-obese women using [^11^C]-Raclopride or the D2 receptor specific radiotracer [^11^C]-NMB [35,37,47]. Similar null-findings have been reported in two samples overlapping with those of Karlsson et al. [38] and Eisenstein et al. [48], as well as in two samples including non-obese individuals only (BMI < 30 kg/m^2^) [39,40].

It is crucial to consider the characteristics of radiotracers when interpreting findings from PET and SPECT studies, in particular in the domain of D2/D3 receptors, as has been discussed extensively before [23]. [^11^C]-Raclopride and [^123^I]-IBZM are similar in their characteristics, with a moderate affinity for D2 receptors and high correlations in binding potential between striatal subregions. These radiotracers are typically used to capture the binding potential of the striatum as a whole. [^18^F]-Fallypride binds to D2/D3 receptors with higher affinity and thus enables the imaging of D2 receptor binding of striatal subregions and extrastriatal regions [61]. In contrast, [^11^C]-(+)-PHNO has a notably higher affinity for D3 receptors. As an agonist, [^11^C]-(+)-PHNO preferably binds to functionally active D2/D3 receptors (high affinity state) rather than the antagonists that bind to both high and low affinity state receptors. D3 receptors are most abundant in the ventral striatum and dopaminergic midbrain. As such, findings localized to these regions tend to be interpreted as D3-receptor-related effects. However, the ventral striatum also expresses D2 receptors abundantly and it has been shown that the [^11^C]-(+)-PHNO binding potential in this region still accounts much stronger for D2 rather than D3 receptor availability. Based on this, authors of a multi-tracer obesity study explain their positive correlation of BMI with [^11^C]-(+)-PHNO binding and the absence of a relationship with the D2-preferring radiotracer [^11^C]-Raclopride as a BMI-related difference in the density of active D2/D3 receptors, rather than of D3 receptors [40]. Because of competition between binding to receptors, the binding potential of these radiotracers can, however, also reflect the amount of endogenous dopamine available in the synaptic cleft. Null-findings from the radiotracer [^11^C]-NMB, which binds specifically to D2 receptors and is not replaceable by endogenous dopamine, tentatively suggest that positive findings with other radiotracers may reflect differences in dopamine levels. We return to this point below (Section 4.1). Future studies implementing [^11^C]-NMB, preferably in a multi-tracer design with a D2/D3 receptor radiotracer that is replaceable by endogenous dopamine, could help to alleviate currently ambiguous findings. Furthermore, a [^11^C]-(+)-PHNO study including a wide BMI range could address the open question if the consistent observations of a positive correlation with BMI reflect a linear relationship that also holds for those with more extreme obesity, or rather a nonlinear relationship.

Regional specificity of differences in D2/D3 receptor binding potential has been observed in some of the reviewed studies that looked into striatal subregions. For example, Guo and colleagues [44] found a positive correlation between BMI and binding potential in the caudate nucleus and putamen and a negative correlation in the ventromedial striatum. Whereas the positive correlation for the caudate nucleus generally concurs with findings from Dunn and colleagues [43], another correlational study reported the opposite, namely a (borderline significant) negative correlation between BMI and binding potential in the caudate nucleus and putamen [45]. Both correlational studies included women and men in a similar BMI range and assessed D2/D3 receptor binding potential using the same radiotracer, [^18^F]-Fallypride [44,45]. Region-specificity of findings is thus unlikely to explain the observed inconsistencies in the domain of D2/D3 receptors in mild to severe obesity.

Age is another important factor to consider when evaluating the reviewed findings [62,63,64]. Dang and colleagues [46] showed that the positive correlation between BMI and D2/D3 receptor binding potential in the striatum depended on age and was only observed for those above the age of 30. Even though most studies controlled for age by adding it as a covariate, the included age range may affect whether or not BMI-related effects can be observed. In the age of no decline (up to 30/35 years old), BMI findings may be less easy to detect because the system is likely more balanced and simply adapts to subtle changes. However, since no clear age-related pattern emerged in this qualitative review, the reported inconsistencies cannot simply be attributed to differences in the included age range.

### 3.2. Dopamine Release

Conclusive evidence for obesity-related differences in dopamine release in the striatum is currently lacking (Table 2). Dopamine release can be investigated using radiotracers that target D2/D3 receptors and compete for binding with endogenous dopamine ([^123^I]-IBZM, [^11^C]-Raclopride, [^18^F]-Fallypride, or [^11^C]-(+)-PHNO). Dopamine release is induced by a certain manipulation and assessed by contrasting the measurement following manipulation to a baseline or placebo measurement. A common manipulation is the administration of a stimulant drug, such as methylphenidate [36] or dexamphetamine [31,45]. Other ways that are thought to stimulate dopamine release are the presentation of food [36,65], or the administration of glucose [33,50].

Drug-induced dopamine release did not vary as a function of BMI [36], nor were differences observed between the obese and a normal-weight control group [31] when a wide range of BMIs was included (up to 60 kg/m^2^). A study in normal-weight to mildly obese participants did observe a moderately strong positive correlation between BMI and dopamine release by dexamphetamine in the right putamen, but only when not correcting for multiple comparisons [45]. Peripheral food stimulation without consumption did not lead to BMI-related differences in dopamine release in a small group of individuals with moderate to severe obesity [36]. When administering either glucose or sucralose orally, the difference in D2/D3 receptor binding potential in the ventral striatum did correlate with BMI in normal-weight to moderately obese individuals, with higher dopamine release (lower binding potential following glucose) for lower BMI and lower dopamine release (higher binding potential following glucose) for higher BMI [50]. Following intravenous administration of a glucose load, Haltia and colleagues [33] observed no difference in dopamine release between a normal-weight and overweight-to-moderately obese group in a notably younger sample. A dopamine release study with [^11^C]-(+)-PHNO measurements in a wide BMI range could provide new insights. Due to its preferred binding to the functional high-affinity state D2/D3 receptors, [^11^C]-(+)-PHNO is particularly sensitive to changes in endogenous dopamine and it can capture changes in both the striatum and midbrain.

Several factors are worth exploring in future work before concluding that dopamine release is not affected in obesity, in particular given the striking discrepancy with positive findings from animal studies in this domain [66,67,68]. It is possible, as speculated by Kessler and colleagues [45], that the relationship between BMI and dopamine release follows an inverted U-shaped curve, with a positive relationship from normal weight to mild obesity (18.5–35 kg/m^2^) and a negative relationship for severe obesity (>40 kg/m^2^). Although this is not supported by secondary analyses of the two studies that sampled the right-hand side of the putative inverted U-shaped curve [31,36], no study to date has set out to investigate this nonlinear relationship explicitly. Second, the exact method should be carefully considered when it comes to drug-induced dopamine release. Wang and colleagues [36] administered a fixed dose of 20 mg of methylphenidate, independent of weight, and observed a negative correlation between BMI and blood plasma levels of methylphenidate. Although plasma levels did not correlate with dopamine release, BMI-dependent uptake of methylphenidate could have masked potential increases in dopamine release (or exacerbated a decrease in release, in line with the hypothesis of an inverted U-shaped relationship, which was not observed). Adjusting the dose to participants’ weight, as done by Kessler and colleagues [45], is thought to address this problem. Without measuring plasma levels of the pharmacological agent, however, this cannot be confirmed. Van de Giessen and colleagues [31] took a similar approach, albeit with a lower dose adjustment (Table 2) based on participants’ “ideal” instead of actual weight. It was indeed shown that plasma levels of dexamphetamine did not differ between the groups [31]. Notably, plasma levels varied strongly between individuals independent of group. Here, the variable drug dose may explain the large variability in measured dopamine release within both groups and, in turn, mask potential group differences. Third, when endogenous dopamine release in response to food-associated stimuli is investigated, it should be carefully considered which signaling pathways contribute to the signal. For example, signals arising from the visual, olfactory and gustatory senses, and signals from the gut via the vagus nerve, contribute to a signal elicited by oral but not intravenous administration of glucose. Finally, it would be worth exploring ways to distinguish between dopamine release in subregions of the ventral striatum in humans in vivo, that is, in the core and shell regions of the nucleus accumbens. Dopamine in these subregions has been suggested to play opposing roles in (mal)adaptive behaviour in rodent studies [66,69,70,71,72].

### 3.3. Dopamine Synthesis

A consistent picture of lower obesity-related dopamine synthesis measures emerges in the range of normal weight up to moderate obesity [51,52,53] (Table 3). The PET radiotracers [^18^F]-FDOPA and [^18^F]-FMT are used to assess aromatic L-amino acid decarboxylase (AADC), an enzyme part of the dopamine synthesis pathways (Figure 1). Two independent studies report a negative correlation between BMI and dopamine synthesis capacity measured in the striatum using [^18^F]-FMT [51,52]. In a multicenter study with a relatively large sample, BMI also correlated negatively with the rate of influx of the radiotracer [^18^F]-FDOPA in the ventral striatum and putamen, which specifically reflects dopamine synthesis capacity [53]. An extensive PET protocol was implemented in the latter study and included a measure of [^18^F]-FDOPA washout rate to calculate the effective distribution volume ratio (EDVR; the ratio of influx to washout) as an estimate of striatal dopamine tone. In line with the [^18^F]-FDOPA influx findings, EDVR was negatively correlated with BMI in all striatal subregions, whereas [^18^F]-FDOPA washout was positively correlated with BMI in the ventral striatum and putamen. These findings suggest that a lower obesity-related dopamine synthesis capacity goes hand in hand with lower striatal dopamine tone, which resonates with the idea that previous D2/D3 receptor-related findings might need to be interpreted in terms of dopamine tone [15] (discussed further below, Section 4.1). A study in non-obese individuals indeed showed a correlation between dopamine synthesis capacity ([^18^F]-FMT) measured in the striatum and D2/D3 receptor binding potential ([^11^C]-Raclopride), but not with methylphenidate-induced dopamine release [73]. However, with an upper BMI limit of 37 kg/m^2^, the three reviewed studies in this domain leave the class of severe and more extreme obesity unsampled. Future work is required to address the question whether or not the positive correlation holds also when severely obese individuals are included.

The consistency of the findings within this domain is noteworthy considering between-study differences in radiotracer ([^18^F]-FMT vs. [^18^F]-FDOPA), sample size (15 vs. 60), sex ratio (~50:50 vs. predominantly men), age range (20–30 vs. 30–42 years old) and inclusion of smokers (none vs. 18 out of 60). Furthermore, different instructions to control food intake prior to scanning were given (Table 3). It has been suggested that dopamine synthesis capacity or dopamine tone is a more stable trait-like measure than D2/D3 receptor availability or dopamine release. and thus less sensitive to subtle differences in either study design or sample characteristics such as age. An interesting question is how stable the measure is, for example, to changes in weight or diet, and if it could in fact capture a predisposition to develop obesity. This should be carefully assessed in future studies.

### 3.4. Dopamine Transporters

Dopamine transporters are responsible for the majority of dopamine reuptake into the presynaptic neuron after release. Measurements of dopamine transporter availability have shown a rather consistent lack of a relationship with degree of obesity [55,56,57,58,59,60] (for a recent review see [25]) (Table 4). Given the substantial sample size in several of these studies, it is unlikely that the absence of an association can be explained by poor statistical power. Also, age and sex were controlled for in most studies. One study with a sample of predominantly normal-weight and overweight individuals did, however, report a negative relationship between dopamine transporter availability and BMI [54].

Differences in the characteristics of the radiotracers used have been brought up as a possible explanation for the discrepancy with findings by Chen and colleagues. The radiotracer [^123^I]-PE2I [56] is said to outperform [^99m^Tc]-TRODAT-1 [54] in imaging quality due to its high selectivity to dopamine (relative to serotonin) transporters, and a higher specific-to-nonspecific-binding ratio. Higher specific binding is also characteristic of [^123^I]-FP-CIT [57,58,59], [^18^F]-FP-CIT [60] and [^123^I]-nor-β-CIT [55]. Because of regional differences in the distribution of dopamine (striatum) and serotonin receptors (midbrain, hypothalamus, thalamus), selectivity of radiotracers may be negligible in the DAT domain.

Another difference between the studies is the degree of overweight of the samples, which was particularly low in the study by Chen and colleagues [54], although not much lower than in two studies presenting null findings [55,60]. Severe obesity has been sampled only sparsely in the DAT domain, with no existing DAT measurements of individuals with a BMI over 50 kg/m^2^. Thus, although there is a wealth of evidence from molecular neuroimaging suggesting no obesity-related differences in DAT in humans (but see [74] for observed post-mortem differences), it is worth investigating the substrate in severe and more extreme obesity.

## 4. Discussion

### 4.1. Integrating Findings across Domains

The available pieces of the puzzle on central dopamine differences in obesity are not readily put together; many pieces do not seem to fit together. Seemingly inconsistent findings within or across the dopamine domains reviewed above might be explained by the existence of a nonlinear relationship between obesity and the respective dopamine target. Several such suggestions have been proposed. Kessler and colleagues [45] speculated that a quadratic relationship between reward sensitivity and degree of obesity may reflect the progression of changes in dopamine release with the development of obesity and decreasing control over eating behavior. Enhanced dopamine release in early overweight and moderate obesity, as observed in their study, could lead to either decreased dopamine release or decreased D2/D3 receptor density in a more severe stage. This resonates with theories of addiction that postulate the transition from initially voluntary to compulsive substance use, paralleled by distinct dopamine alterations [75,76,77], although it is highly debated whether or not obesity can be understood in the framework of addiction [78,79,80,81]. The existence of a nonlinear relationship with BMI was also proposed for D2/D3 receptor binding potential in a meta-analysis of [^11^C]-Raclopride studies [37] and a qualitative review of group studies assessing D2/D3 receptor binding potential with a variety of radiotracers [15]. Based on radiotracer characteristics, the relationship was argued to reflect a quadratic relationship of opposite sign with tonic extracellular dopamine levels, with lower dopamine tone in overweight to moderate obesity and higher tone in severe obesity [15]. The left-hand side of the parabola has recently been confirmed in a reviewed [^18^F]-FDOPA study [53], whereas the right-hand side has not yet been investigated.

Radiotracers that bind to D2/D3 receptors, except for [^11^C]-NMB, are replaceable by endogenous dopamine and therefore compete for binding with these receptors. As such, the measured binding potential is affected by both the density of receptors, as well as the amount of dopamine present in the synapse. Given these characteristics, it is conceivable that the quadratic relationship between BMI and D2/D3 receptor binding potential rather captures an interaction between receptor availability and dopamine released in the synapse, as suggested in a more comprehensive review of the literature that covers findings in the domains of D2/D3 receptors, dopamine release, and dopamine transporters [23]. An interaction of this kind is biologically plausible and can explain the somewhat puzzling “tipping point” of the parabola. According to the hypothesized parabola, at a given degree of severe obesity, D2/D3 receptor binding potential has to be indistinguishable from that of individuals in the normal weight range. It is unlikely that, in the progression from overweight to severe obesity, an individual goes from suboptimally high (low) to suboptimally low (high) D2/D3 receptor availability (dopamine tone). Compensatory downregulation of D2/D3 receptors in response to an increases in dopamine tone, or vice versa, could mask obesity-related differences as measured with D2/D3 receptor radiotracers that are replaceable by endogenous dopamine. Although an interaction account is appealing, many pieces of the puzzle for determining its exact nature are missing. Multi-tracer studies, e.g., assessing dopamine synthesis capacity and D2(/D3) receptor availability in the same individuals and ideally including a wide BMI range, could provide additional answers.

An alternative, and not mutually exclusive, interpretation is that the putative nonlinear relationships reflect multiple ways to obesity. Obesity is a heterogeneous condition with multifactorial etiology. For some, obesity results from compulsive binge-eating episodes, whereas others consistently consume more moderate amounts of excessive energy. The intake of dietary components that are known to affect the dopamine system, such as saturated fats and sugar, may also differ greatly between individuals. The studies reviewed here may have unintentionally tapped into different obesity subpopulations that are not characterized by the same impairment of the dopamine system-interaction or in a single substrate. In fact, some cases of obesity may not be accompanied at all by dopamine differences. Synthesizing findings from different samples as if they are part of the same population could lead to a distorted picture. It will be important for future PET/SPECT studies to characterize participants in detail and report on these characteristics transparently and systematically, in line with recommendations for good scientific practice proposed for food-related neuroimaging studies [82]. This will allow for a more meaningful synthesis of research findings in qualitative or quantitative reviews, as well as in secondary analysis of existing data, i.e., mega-analysis.

### 4.2. Limitations

Despite the great added value of PET and SPECT studies for approximating molecular targets in humans in vivo, the methods used are inherently limited. Due to their binding characteristics, most radiotracers are highly selective, binding with high affinity to only one or a few molecular substrates in specific regions of the brain. At the same time, the measurements of a given radiotracer do not allow us to quantify parts of the dopamine pathways unequivocally. As discussed by others [23] this complicates the interpretation of PET and SPECT findings and requires careful consideration of the characteristics of the radiotracer used in each study. Due to the regional specificity of many dopaminergic radiotracers, the focus of this review is on studies that reported findings from striatal and midbrain dopamine measurements. It is difficult at present to assess alterations in extrastriatal and/or dopaminergic midbrain regions of the brain in humans in vivo. This is problematic, because the dopamine system includes functionally relevant connections with many other brain regions such as the prefrontal cortex, hypothalamus, hippocampus, and amygdala. In the case of a dopamine anomaly, it is likely that extrastriatal regions contribute to the development and expression of cognitive symptoms. The one study that was excluded from this review used a radiotracer targeting extrastriatal D2/D3 receptors ([^11^C]-FLB 457) and found a positive correlation with BMI in the amygdala in a nonobese sample [29]. Another multi-tracer PET study found that D2/D3 receptor availability in severely obese individuals correlated positively with a measure of brain glucose metabolism in the prefrontal cortex [34]. This was not the case for non-obese controls, despite significantly higher D2/D3 receptor availability in this group. Studies performing extrastriatal measurements of dopamine targets or prefrontal glucose metabolism, ideally as part of multi-tracer study designs including striatal and midbrain measurements, could give us a more complete picture of central dopamine imbalances in obesity.

The dopamine system is, of course, an integral part of the brain and the body as a whole. Of note in this context is that dopamine interacts with other neurotransmitter systems that have also been associated with obesity, such as the serotonin (reviewed in [23,55,58,59]) and opiate system [37,38,83], as well as with the neuroendocrine hormone insulin [43,84,85,86,87,88]. PET and SPECT studies targeting multiple neurotransmitter systems in the same sample may provide clearer answers in this regard. This approach has already led to valuable insights suggesting that the interaction between dopamine and opiates plays a role in obesity [38] and that the impaired interaction can be reversed by weight loss after bariatric surgery [83]. Whether or not interactions between neurotransmitter systems could give rise to the proposed nonlinear relationships is an open question that warrants further investigation. Furthermore, the promise of a combined fMRI-PET approach was recently demonstrated in an elegant study, which showed that decreased dopamine release in the ventral striatum was accompanied by enhanced functional connectivity between the ventral striatum and hypothalamus, ventral striatum and VTA, and VTA and prefrontal cortex 45 min after intranasal insulin administration in healthy, normal-weight young men [88]. 

For the development of effective strategies for prevention and treatment, the question arises which central dopamine difference, be it cause or consequence, can be modulated to rebalance the system—and how. An answer is not possible based on the cross-sectional nature of the studies reviewed here. Only a few dopamine PET or SPECT studies assessed the effects of weight loss, mostly following bariatric surgery (reviewed in [24]). Longitudinal and intervention studies are required to understand and disentangle the causal effects of changes in adiposity, diet, physical (in)activity, and other relevant factors. Furthermore, medications that act on central neurotransmitter systems, including dopamine, present an opportunity to investigate the rebalancing of the dopamine system in more detail. The two most widely prescribed antiobesity medications in the USA, phentermine and bupropion/naltrexone, stimulate dopamine, among others [89,90]. To our knowledge, PET or SPECT studies with measurements before and after prescription of these medications have not yet been conducted. Subtle pharmacological manipulations with other dopaminergic agents, for example, a D2/D3 receptor agonist or antagonist, could also be used to study delicate obesity-related interactions in the dopamine system and beyond.

A major limitation of the current review, and the field in general, is the operationalization of obesity in terms of distinct BMI categories. Although BMI categories are well-defined and commonly used to determine adiposity and related health risks, it is also widely criticized for use as a single measure because of its heterogeneous nature [91,92,93]. This may explain the absence of significant linear correlations between BMI and dopamine measurements in many studies, while group means do differ due to higher discriminative power. We need to appreciate the multifactorial nature of obesity and better characterize individuals’ obesity status. Including additional measures of body shape, lean and fat mass, energy expenditure, metabolism, obesity onset and duration, diet, and physical activity as a default could help identify different obesity profiles. It should be stressed, too, that the cutoff for extreme obesity as introduced in this review is arbitrary and by no means represents a “new” category. Nevertheless, it may be relevant, based on the observed differences in D2/D3 receptor binding potential, to think of a further classification of the category of severe obesity, which is much broader than any other BMI category and seems to represent a heterogeneous group.

Finally, we emphasize the qualitative nature of the current review. Although the literature search closely follows existing guidelines for systematic reviews of non-intervention studies [28], not all precautions could be taken to completely rule out bias in the reporting of findings. The literature search, selection of articles and extraction of reported findings was performed by a single researcher. The review was not pre-registered because it served as background research for future work. Further, to maximize the knowledge gained from as many valuable PET and SPECT studies as possible, results from primary as well as secondary analyses are included in the review. That is, when a study focused on dopamine release as a primary outcome, and baseline differences were also reported as a secondary outcome, both were included. Additionally, where studies reported secondary findings for obese individuals that served as a control group for, for example, a group with binge-eating disorder, these secondary (but not primary) findings were included. It was not always straightforward to judge the robustness or risk of bias of the results, especially when no mention is made of (1) correction for multiple comparisons, (2) what primary and secondary outcomes are, and (3) what data have already been published. An extensive quantitative review with risk of publication bias assessment (meta-analysis) and a secondary analysis of pooled data (mega-analysis) across dopamine domains is recommended to address these issues.

## 5. Conclusions

Based on a wealth of evidence, a role for central dopamine differences in obesity seems undeniable. Currently, the evidence leans towards lower striatal D2/D3 receptor binding potential in extreme obesity and higher binding potential in overweight and moderate obesity, which could reflect differences in D2/D3 receptor availability, dopamine tone, or a combination of the two. Overweight to moderate obesity may also be characterized by a higher dopamine synthesis capacity in the striatum. Furthermore, obesity-related differences in dopamine release or transporter availability have not convincingly been shown in humans. The existence of a nonlinear relationship between severity of obesity and dopamine differences have been proposed by several researchers to explain seemingly inconsistent findings either within or between dopamine domains. Whether or not such nonlinear relationships exist, and for what parts of the dopamine pathways, remains to be tested. How to explain the existence of the proposed nonlinear relationships is also an important open question. Does it reflect a natural progression in the development of moderate to severe obesity? Is it the result of an interaction between differences in D2/D3 receptor density and tonic dopamine levels? And could it be that a nonlinear relationship captures distinct (dopamine) anomalies for different obesity profiles? Despite many observed associations between obesity and components of the dopamine system, it is unlikely that obesity can be traced back to a single dopaminergic cause or consequence. For effective personalized prevention and treatment of obesity, it will be crucial to identify possible dopamine (and non-dopamine) profiles and their functional characteristics. To identify helpful dopamine targets for different stages of obesity (development), large multi-tracer and multi-modal neuroimaging studies are required that better characterize overweight and obesity in terms of body shape, lean and fat mass, energy expenditure, metabolism, obesity onset and duration, diet, and physical activity, and can identify different obesity profiles. Longitudinal and intervention studies (e.g., diet, weight loss, dopamine manipulation) are crucial for understanding the progression of dopaminergic changes that leads to an imbalance of the system, and the targets that can be used to reverse this imbalance at different stages.

## Figures and Tables

**Figure 1 brainsci-12-00486-f001:**
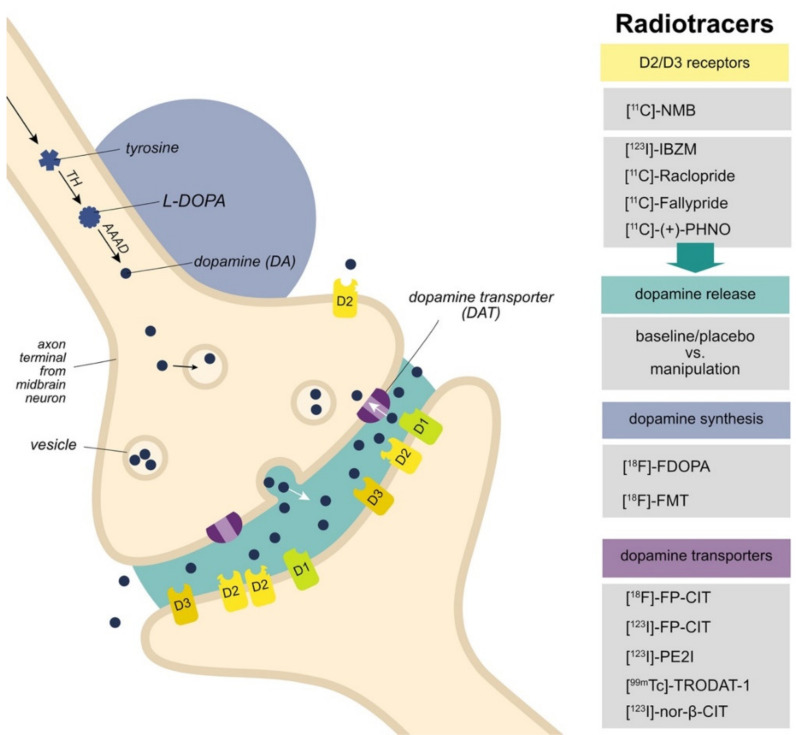
Dopamine substrates that are commonly quantified in humans in vivo using PET/SPECT. A schematic representation of a dopaminergic synapse at an axon terminal in the striatum. The soma of the neuron projecting to the striatum is located in the midbrain. The last two steps of the dopamine synthesis pathway are depicted (blue): tyrosine hydroxylase (TH) converts tyrosine to L-DOPA, which is then converted to dopamine by aromatic L-amino acid decarboxylase (AAAD). Dopamine is packed in vesicles for both tonic and phasic release. Phasic release (green) is triggered by action potentials and can be stimulated by a challenge such as amphetamine, food intake, or food stimuli. Dopamine transmits signals from the pre- to the postsynaptic neuron by binding to postsynaptic receptors (yellow) and activating a cascade of events in the postsynaptic neuron. D1, D2 and D3 receptors are depicted. Due to the nature of most radiotracers that target dopamine receptors, i.e., they compete for binding with endogenous dopamine, dopamine release is derived from contrasting binding of the tracer before and after a dopamine challenge. After release, dopamine is transported back into the presynaptic neuron by means of dopamine transporters (DAT, purple). For each domain, common radiotracers are listed on the right. [^11^C]-NMB: Carbon-11-labeled N-methyl benperidol, [^123^I]-IBZM: Iodine-123-labeled iodobenzamide, [^11^C]-Raclopride: Carbon-11-labeled raclopride, [^11^C]-(+)-PHNO: Carbon-11-labeled (+)-4-propyl-9-hydroxynaphthoxazine, [^18^F]-FDOPA: Fluorine-18-labeled L-dihydroxyphenylalanine, [^18^F]-FMT: Fluorine-18-labeled Fluoro-L-m-tyrosine, [^18^F]-/[^123^I]-FP-CIT: Fluorine-18-labeled/Iodine-123-labeled ioflupane, [^123^I]-PE21: Iodine-123-labeled N-(3-iodoprop-2E-enyl)-2-b-carbomethoxy-3b-(4-methylphenyl) nortropane, [^99m^Tc]-TRODAT-1: Technetium-99m-labeled Tropane for imaging Dopamine Transporters, [^123^I]-nor-β-CIT: Iodine-123-labeled 2beta-carbomethoxy-3beta-(4-iodophenyl)nortropane.

**Figure 2 brainsci-12-00486-f002:**
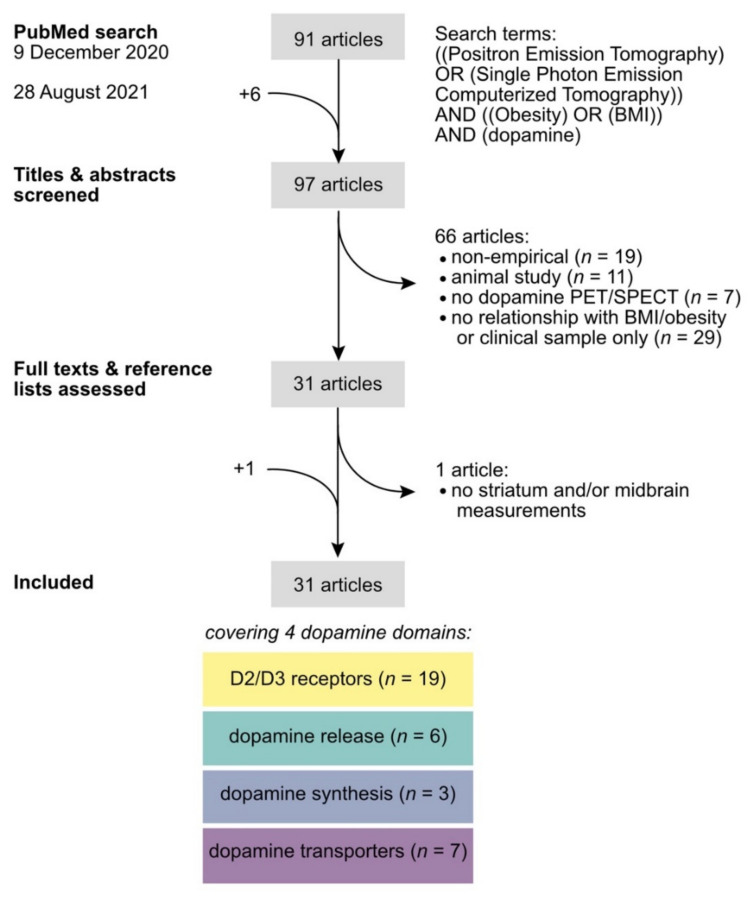
Flow chart of the literature search. Grey boxes highlight the number of hits for each step. Colored boxes contain the number of records of which relevant findings were reported in the retrieved articles per dopamine domain.

**Table 1 brainsci-12-00486-t001:** D2/D3 receptor studies. Imaging details, sample characteristics, and BMI/obesity-related findings are grouped by imaging method (SPECT/PET), reference and radiotracer. The radiotracer, type of quantification of in vivo measurements (dependent variable, DV) and the analyzed brain regions of interest (ROI) and/or whole-brain analysis (WB) are reported. DV include: BP_ND_ = non-displaceable binding potential, DVR = distribution volume ratio; ROI include: STR = striatum, CAU = caudate nucleus, PUT = putamen, PAL = pallidum, MB = dopaminergic midbrain (Substantia Nigra (SN) and/or ventral tegmental area (VTA)) and can be specified by prefixes “v” = ventral, “vm” = ventromedial, “d” = dorsal, “a” = anterior, “p” = posterior. Sample size is given as the number of females and males (F:M) per group. Upon availability of the data, the sample is characterized in terms of BMI group(s) normal-weight (NW), overweight (OW) or obese (OB), including mean BMI with standard deviation (M(SD), or confidence intervals, CI) and range, mean age with standard deviation (M(SD), or confidence intervals, CI) and range, fasting/fed state and smoking status of included participants. Values with a ± sign reflect an estimation based on the available data.

Reference	Imaging			Sample								Primary/ Secondary Analysis	Findings
				N	BMI			Age		Fasting	Smoking		
	Radiotracer	DV	WB/ROI	F:M	Groups	M(SD)	Range	M(SD)	Range				
SPECT studies													
De Weijer et al., 2011 [30]	[^123^I]-IBZM	D2/D3 BP_ND_	STR	15:0	NW + OW	21.7(2.1)	19.5–27.6	28(10.4)	20–60	Not fasted, not fed		Primary	OB < NW + OW BMI: N.S. in separate groups
				15:0	OB	46.8(6.5)	38.7–61.3	37.8(7)	26–49	Fasted (overnight)			
van de Giessen et al., 2014 [31]	[^123^I]-IBZM (+baseline)	D2/D3 BP_ND_	STR	15:0	NW	21.8(1.8)	18.5–24.9	38.5(5.6)	18–45	Fed (standard breakfast)	None	Secondary	OB < NW BMI: N.S. in separate groups
				15:0	OB	42.9(4.9)	36.3–56.6	36.3(4.0)	18–45				
PET studies													
Wang et al., 2001 [32]	[^11^C]-Raclopride	D2/D3 DVR = BP_ND_ + 1	STR	3:7	NW + OW	24.7(2.6)	21–28	37.5(5.9)	25–54		1 light smoker	Primary ^1^	OB < NW + OW BMI (OB): STR↓
				5:5	OB	51.2(4.8)	42–60	38.9(7.3)	26–54		None		
Haltia et al., 2007 [33]	[^11^C]-Raclopride (+placebo)	D2/D3 BP_ND_	WB; CAU, PUT	6:6	NW	22(1)	<24	±25.5(4.5)		Fasted (overnight)	No heavy smokers	Secondary	OB < NW in STR (WB only); BMI (OW/OB): PUT↓
				6:6	OW + OB	33(5)	>27	±25(2.5)					
Volkow et al., 2008 [34]	[^11^C]-Raclopride	D2/D3 DVR = BP_ND_ + 1	STR (CAU + PUT)	6:6	NW + OW	25(3)	<30	33.2(8)	20–55		None	Secondary ^1^	OB < NW + OW
				5:5	OB	51(5)	>40	35.9(10)	20–55				
Steele et al., 2010 [35]	[^11^C]-Raclopride	D2/D3 DVR = BP_ND_ + 1	a, pCAU a, pPUT, vSTR	5:0	NW	21.3		21.8			No tobacco use >once a week	Primary	OB = NW
				5:0	OB	44.8	40–53	32.2	20–38				
Wang et al., 2011 [36]	[^11^C]-Raclopride (+placebo)	D2/D3 BP_ND_	CAU, PUT, vSTR	5:3	OB	36.5(9.4)	31–59	41.8(8.9)	28–56	Fasted (overnight)	No nicotine use >1 pack/day	Secondary	BMI: N.S.
Karlsson et al., 2015 [37]	[^11^C]-Raclopride	D2/D3 BP_ND_	WB + dCAU, PUT, vSTR	14:0	NW	22.7(2.9)		44.9(12.9)		Fasted (2 h before scan)	None	Primary ^2^	OB = NW
				13:0	OB	41.9(3.9)	37.1–49.3	39.1(10.7)			5		
Tuominen et al., 2015 [38]	[^11^C]-Raclopride	D2/D3 BP_ND_	CAU, PUT, vSTR	20:0	NW	22.4(2.6)		42.0(13.2)		Fasted (2 h before scan)	None	Secondary ^2^	OB = NW
				25:0	OB	41.3(4.1)		41.2(9.2)			8		
Cho et al., 2015 [39]	[^11^C]-Raclopride	D2/D3 BP_ND_	dPUT, dCAU, vSTR	0:25	NW-OW	22.0(2.5)	17.6–28.0	23.3(2.9)	18–29		5	Primary	BMI: N.S. (Asymmetry Index in dPUT↑)
Caravaggio et al., 2015 [40]	[^11^C]-Raclopride	D2/D3 BP_ND_	dCAU, dPUT, vSTR	14:21	NW-OW	23.2(2.7)	18.6–27.8	31.3(9.0)	20–47	Not fasted	None	Primary	BMI: N.S.
	[^11^C]-(+)-PHNO	D2/D3 BP_ND_	dCAU, dPUT, vSTR	7:19	NW-OW	23.7(3.0)	18.6–27.8	29.9(7.4)	20–45	Not fasted	None	Primary	BMI: vSTR↑
Cosgrove et al., 2015[41]	[^11^C]-(+)-PHNO	D2/D3 BP_ND_	dCAU	2:10	NW-OB		20.8–36.5		19–37		None	Secondary	BMI: right dCAU↑
Gaiser et al., 2016 [42]	[^11^C]-(+)-PHNO	D2/D3 BP_ND_	CAU, PAL, PUT, MB (SN/ VTA), vSTR	4:10	NW	22.3(1.8)	18.5–24.9	34.9(10.2)		Fasted (overnight)	None	Primary	OB > NW in vSTR, PAL, MB BMI (with OW): vSTR, PAL, MB↑
				1:13	OW	27.2 (1.3)	25.0–29.9	36.7(11.5)			1		
				4:10	OB	35.5(4.5)	>29.9	37.0(10.1)			None		
Dunn et al., 2012 [43]	[^18^F]-Fallypride	D2/D3 BP_ND_	CAU, PUT, vSTR, SN	8:0	NW	23(2)	<25	40(9)		Fasted (8 h before scanning)	None	Secondary	OB > NW in CAU
				14:0	OB	40(5)	>30	40(8)					
Guo et al., 2014 [44]	[^18^F]-Fallypride	D2/D3 BP_ND_	WB + CAU, PUT, vmSTR	11:12	NW + OW	22.4 CI (21.3,23.5)	>18	28 CI (25.1, 30.4)	18–45	Fed (standardized breakfast 2 h before scan)	None	Primary	BMI: dSTR↑; vmSTR↓
				10:10	OB	36.1 CI (34.9,38.3)	<45	35 CI (31.9, 38.8)	18–45				
Kessler et al., 2014 [45]	[^18^F]-Fallypride (baseline)	D2/D3 BP_ND_	CAU, PUT, vSTR, SN/vMB	15:18	NW-OB	24.8	19–35	25.8	18–35			Primary	BMI: CAU↓(borderline significant)
Dang et al., 2017 [46]	[^18^F]-Fallypride	D2/D3 BP_ND_	WB + CAU, PUT, vSTR, MB	72:58	NW-OB	25.5(4.8)	18.5–40	35.6(18.2)	18–81		None	Primary	BMI (controlled for age and gender): PUT↑, (above 30 years old only): all ROI↑
Eisenstein et al., 2013 [47]	[^11^C]-NMB	D2 BP_ND_	WB + CAU, PUT, vSTR	11:4	NW + OW	22.6(2.2)	18.9–27.7	29.7(5.6)	22–40	Not fasted, not fed	None	Primary ^3^	OB = NW + OW, BMI: N.S.
				12:3	OB	40.3(4.9)	33.2–47.0	32.5(5.9)	25–41				
Eisenstein et al., 2015 [48]	[^11^C]-NMB	D2 BP_ND_	dSTR (CAU + PUT), vSTR (Nac), MB	13:4	NW + OW	22.1(2.0)	18.7–25.9	28.5(5.5)	21–39	Not fasted, not fed	None	Secondary ^3^	OB = NW + OW, BMI: N.S.
				19:3	OB	39.6(5.2)	33.4–51.0	31.4(6.3)	23–40				

^1^ largely overlapping obese samples. ^2^ overlapping obese samples. ^3^ overlapping samples.

**Table 2 brainsci-12-00486-t002:** Dopamine release studies. Imaging details, sample characteristics, and BMI/obesity-related findings are grouped by imaging method (SPECT/PET), reference and radiotracer. The radiotracer, type of quantification of in vivo measurements (dependent variable, DV) with manipulation to induce dopamine release and the analyzed brain regions of interest (ROI) and/or whole-brain analysis (WB) are reported. DV include: ΔBP_ND_ = difference in non-displaceable binding potential; ROI include: STR = striatum, vSTR = ventral striatum, CAU = caudate nucleus, PUT = putamen, SN = substantia nigra. Sample size is given as the number of females and males (F:M) per group, if not otherwise stated. Upon availability of the data, the sample is characterized in terms of BMI group(s) normal-weight (NW), overweight (OW) or obese (OB), including mean BMI with standard deviation (M(SD)) and range, mean age with standard deviation (M(SD)) and range, fasting/fed state and smoking status of included participants. Values with a ± sign reflect an estimation based on the available data. * Same sample as Haltia et al. (2007) [33] with additional expected glucose measurements.

Reference	Imaging			Sample								Primary/Secondary Analysis	Findings
				N	BMI			Age		Fasting	Smoking		
	Radiotracer	DV + manipulation	ROI	F:M	Groups	M(SD)	Range	M(SD)	Range				
SPECT studies													
van de Giessen et al., 2014 [31]	[^123^I]-IBZM	ΔD2/D3 BP_ND_ *0.3 mg/kg d-amphetamine* *vs. baseline (oral)*	STR	15:0	NW	21.8(1.8)	18.5–24.9	38.5(5.6)	18–45	Fed (standard breakfast)	None	Primary	OB = NW
				15:0	OB	42.9(4.9)	36.3–56.6	36.3(4.0)	18–45				
PET studies													
Haltia et al., 2007 [33]	[^11^C]-Raclopride	ΔD2/D3 BP_ND_ *300 mg/kg glucose* vs. *placebo (intravenous)*	WB; CAU, PUT	6:6	NW	22(1)	<24	±25.5(4.5)		Fasted (overnight)	No heavy smokers	Primary	OW + OB = NW
				6:6	OW + OB	33(5)	>27	±25(2.5)					
Haltia et al., 2008 [49]	[^11^C]-Raclopride	ΔD2/D3 BP_ND_ *expected glucose (placebo)* vs. *placebo (intravenous) **	WB; CAU, PUT, vSTR	6:6	NW	22(1)	<24	±25.5(4.5)		Fasted (overnight)	No heavy smokers	Primary	OW + OB = NW
				6:6	OW + OB	33(5)	>27	±25(2.5)					
Wang et al., 2011 [36]	[^11^C]-Raclopride	ΔD2/D3 BP_ND_ *20 mg. methylphenidate* vs. *placebo (oral)*	CAU, PUT, vSTR	5:3	OB	35.5(9.4)	31–59	41.8(8.9)	28–56	Fasted (overnight)	No nicotine use >1 pack/day	Secondary	BMI: N.S. (but interaction between MPH and food stimulation in BED group)
		ΔD2/D3 BP_ND_ *food* vs. *neutral stimulation (view, smell, taste (not eat))*											
Wang et al., 2014 [50]	[^11^C]-Raclopride	ΔD2/D3 BP_ND_ *75 g glucose drink* vs. *sucralose drink (oral)*	WB; vSTR	19 F + M	NW-OB		21–35		40–60	Fasted (overnight)	No nicotine dependence	Primary	BMI: vSTR↑, with DA increase (decrease) for lower (higher) BMI
Kessler et al., 2014 [45]	[^18^F]-Fallypride	ΔD2/D3 BP_ND_ *0.42 mg/kg d-amphetamine* vs. *baseline (oral)*	CAU, PUT, vSTR, SN	8:8	NW-OB	25.2	19–35	24.3	21–32				BMI: rPUT, lSN↑

**Table 3 brainsci-12-00486-t003:** Dopamine synthesis capacity studies. PET imaging details, sample characteristics, and BMI/obesity-related findings are grouped by reference and radiotracer. The radiotracer, type of quantification of in vivo measurements (dependent variable, DV) and the analyzed brain regions of interest (ROI) are reported. DV include: K_i_ = net influx rate, EDVR = effective distribution volume ratio, k_occ_ = influx rate constant, k_loss_ = washout rate; ROI include: STR = striatum, CAU = caudate nucleus, PUT = putamen, and can be specified by prefixes “v” = ventral and “d” = dorsal. Sample size is given as the number of females and males (F:M) per group. Upon availability of the data, the sample is characterized in terms of BMI group(s) normal-weight (NW), overweight (OW) or obese (OB), including mean BMI with standard deviation (M(SD)) and range, mean age with standard deviation (M(SD)) and range, fasting/fed state and smoking status of included participants. Values with a ± sign reflect an estimation based on the available data.

Reference	Imaging			Sample								Primary/Secondary Analysis	Findings
				N	BMI			Age		Fasting	Smoking		
	Radiotracer	DV	ROI	F:M	Groups	M(SD)	Range	M(SD)	Range				
Wilcox et al., 2010 [51]	[^18^F]-FMT	K_i_	dCAU, dPUT, vSTR	9:6	NW-OB	25.3	±19.0–33.0	22.9	20–30		No regular smokers	Primary	BMI: dCAU↓
Wallace et al., 2014 [52]	[^18^F]-FMT	K_i_	dSTR (CAU)	8:8	NW-OB		20.2–33.4		20–30	Fed (light meal before scan)		Primary	BMI: rCAU↓
Lee et al., 2018 [53]	[^18^F]-FDOPA	EDVR = k_occ/_k_loss_	CAU, PUT, vSTR	11:49	NW-OB	25.2(3.3)	19.2–36.6	36.4(3.8)	30–43	Fed (no protein-containing foods on scan day)	n = 18	Primary	EDVR: BMI↓ (all ROI) k_occ_: BMI↓ (vSTR + PUT) k_loss_: BMI ↑(vSTR + PUT)

**Table 4 brainsci-12-00486-t004:** Dopamine transporter studies. Imaging details, sample characteristics, and BMI/obesity-related findings are grouped by imaging method (SPECT/PET), reference and radiotracer. The radiotracer, type of quantification of in vivo measurements (dependent variable, DV) and the analyzed brain regions of interest (ROI) are reported. REF = Reference region. DV include: BP_(ND)_ = (nondisplaceable) binding potential; ROI include: STR = striatum, vSTR = ventral striatum, CAU = caudate nucleus, PUT = putamen. Sample size is given as the number of females and males (F:M) per group. Upon availability of the data, the sample is characterized in terms of BMI group(s) normal-weight (NW), overweight (OW) or obese (OB), including mean BMI with standard deviation (M(SD)) and range, mean age with standard deviation (M(SD)) and range, fasting/fed state and smoking status of included participants.

Reference	Imaging			Sample								Primary/Secondary Analysis	Findings
				N	BMI			Age		Fasting	Smoking		
	Radiotracer	DV	ROI	F:M	Groups	M(SD)	Range	M(SD)	Range				
SPECT studies													
Chen et al., 2008 [54]	[^99m^Tc]- TRODAT-1	ROI/ REF	STR	27:23	NW-OB	23.0	18.7–30.6	30.3	20–57		No exclusion criterion	Primary	BMI: STR↓
Koskela et al., 2008 [55]	[^123^I]-nor-β-CIT	BP	STR	15:16	NW-OB	25.6(3.5)	19.1–31.9	25.4(1.3)	24–27	Fasted (overnight)	2	Primary	BMI: N.S. Low vs. high BMI twins: N.S.
Thomsen et al., 2013 [56]	[^123^I]-PE2I	BP	STR (CAU + PUT)	6:6	NW	22.7 (1.4)	21–24.5	48(13.9)	28–69			Primary	OB = OW = NW BMI: N.S.
				4:5	OW	26.9 (1.6)	25.3–29.7	59.8(8.2)	46–71				
				6:6	OB	38.5(5.7)	30.9–49.5	44.3(12.3)	29–59				
van de Giessen et al., 2013 [57]	[^123^I]-FP-CIT	BP	STR (CAU + PUT)	56:67	NW-OB	25.2(3.8)	18.2–41.1	53.3(18.3)	20–83	Not fasted, not fed		Primary	BMI: N.S.
Versteeg et al., 2016 [58]	[^123^I]-FP-CIT	BP	STR	8:0	NW	21.3(1.3)		30.9 (10.5)		Fasted (overnight)	None	Secondary	OB = NW BMI: N.S.
				10:2	OB	36.6(4.4)		31.7(8.9)					
Nam et al., 2018 [59]	[^123^I]-FP-CIT	BP	STR (CAU + PUT)	51:91	NW-OW	25.1(2.8)		61.9(11.4)	>30			Primary	BMI: N.S.
				13:27	OB	32.9(3)		58.2(10.1)	>30				
PET studies													
Pak et al., 2020 [60]	[^18^F]-FP-CIT	BP_ND_ (placebo)	PUT, CAU, vSTR	0: 33	NW-OB	23.1(2.2)	<35	24.5(2.8)	20–31	Fasted (overnight)	No heavy smokers	Primary	BMI: N.S.
		ΔBP_ND_ vs. 300 mg/kg glucose										Primary	BMI: N.S. (BMI: glucose BP_ND_ in vSTR↓)

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
