# Peer review of "Molecular Imaging of Central Dopamine in Obesity: A Qualitative Review across Substrates and Radiotracers"

_brainsci, 2022, doi:10.3390/brainsci12040486_

Round 1
Reviewer 1 Report
The current article represents a literature review focussed on the relationship between BMI/weight status and dopamine function in striatal and midbrain regions in human subjects. The literature search was focused on four domains of dopamine function: D2/D3 receptors, dopamine release, dopamine synthesis, and dopamine transporters. The main finding is that althought the is a clear involvement of the dopamine system in the obese state, it is not likely that there is a causal relationship between dopamine function and obesity.
The authors did a fantastic job summarizing the data and providing a wider perspective on individual findings.
Author Response
The authors would like to thank the reviewer for expressing their appreciation of the work presented in the current article. As no points for improvement were given, no response letter was uploaded.
Reviewer 2 Report
This manuscript by Janssen and Horstmann seeks to analyze the relationship between dopamine and obesity, and use human neuroimaging papers as the source of their meta-analysis. Specifically, they examine dopamine receptors, synthesis, release, and transporters. Overall the review succinctly identifies current limits on our understanding of dopamine in obese and non-obese individuals, and will be a valuable resource for developing new studies.
Major considerations
- The transparency and selectivity of the meta-analysis was well described and was specific to the topic of interest.
- The authors note that one limitation of most imaging studies is the difficulty associated with getting cohort data. I would be curious to know whether any of these studies recorded the approximate length of time that an individual had been obese (e.g., 2 years, versus 15), and whether the duration of time that an individual is obese explains some of the variability in measured changes in dopamine (e.g., D2/D3 binding in mild/severe individuals). This could also be relevant for why general age seems to be a poor explanatory variable.
- The failure to detect changes in dopamine release is quite striking, especially considering how much non-human literature suggests that there should be. The final point raised by the authors, that differences in multiplicative food sensory inputs versus intravenous glucose infusion may account for some of these null reports is quite interesting.
- Do any of the studies related to dopamine release provide greater anatomical precision beyond “ventral” or “dorsal” striatum? Dopamine’s role in striatal subregions have been shown to be quite different, and sometimes oppositional, to nearby subregions; could this account for some of the null results?
Minor considerations
- It is interesting that the most consistent findings arose from the least well studied tracer (dopamine synthesis). The idea that it could be used to capture states/stable traits of dopamine systems is very interesting.
- Have multitracers studies with other neurochemicals been attempted? Successful? It would be interesting to have a real example of this. This comment is specifically related to the Discussion on teasing apart tracers that interact with dopamine vs dopamine receptors (~line 357).
- Thank you for including a brief discussion on the problems with BMI as a proxy for obesity.
